# Distribution of aerophilous diatom communities associated with terrestrial green macroalgae in the South Shetland Islands, Maritime Antarctica

**Juliana Ferreira da Silva**[1]☯, **Maria Angélica Oliveira Linton**[2]☯, **Raylane Ribeiro da Anunciação**[3]‡, **Eduardo Pereira da Silva**[1]‡, **Rodrigo Paidano Alves**[4]☯, **Adriano Luis Schünemann**[1]‡, **Filipe de Carvalho Victoria**[1]☯*, **Margéli Pereira de Albuquerque**[1,5]‡, **Antônio Batista Pereira**[1]☯

1 Antarctic Vegetation Studies Center, Federal University of Pampa, São Gabriel, Rio Grande do Sul, Brazil, 2 Laboratory of Phycology, Federal University of Santa Maria, Santa Maria, Rio Grande do Sul, Brazil, 3 Laboratory of Phycology, Federal Rural University of Pernambuco, Recife, Pernambuco, Brazil, 4 Max Planck Institute for Chemistry, Manaus, Amazonas, Brazil, 5 National Institute of Antarctic Science and Tecnology, Federal University of Pampa, São Gabriel, Rio Grande do Sul, Brazil

☯ These authors contributed equally to this work.
‡ These authors made smaller but essential contributions to this work.
* filipevictoria@unipampa.edu.br

**Data Availability Statement:** All relevant data are within the manuscript and its Supporting Information files. The protocols used in the

## Abstract

The establishment of diatom communities depends on environmental factors such as the type of substrate and geographic conditions that influence the dispersal processes of these organisms. The main goal of this study was to evaluate the similarity between diatom communities associated with the macroalgae *Prasiola crispa* (Lightfoot) Kützing in relation to spatial distance from six sampled sites located in the South Shetland Islands, Maritime Antarctica. The diatom flora associated with *Prasiola crispa* was represented by 23 species distributed in 15 genera. *Pinnularia australoschoenfelderi* Zidarova, Kopalová & Van de Vijver, *Luticola austroatlantica* Van de Vijver, Kopalová, S.A.Spaulding & Esposito, *Luticola amoena* Van der Vijver, Kopalová, Zidarova & Levkov, *Pinnularia austroshetlandica* (Carlson) Cleve-Euler and *Psammothidium papilio* (D.E. Kellogg et al.) Kopalová & Zidarova were the most abundant species in our samples, together they represented 68% of the total number of individuals collected. There was great similarity and abundance of the diatom communites among the sampled points, which resulted in the absence of a linear relationship pattern with distance between sampling points. We conclude that distance was not a factor of differentiation of Antarctic diatom communities associated with terrestrial green macroalgae. This suggests that Antarctic environments may have unique characteristics with homogeneous abiotic factors, at least in relation to this substrate.

presented work are submitted to protocols.io platform.

**Funding:** This work are funded by the National Council for Research and Development (CNPq; process no. 574018/2008) and the Research Foundation of the State of Rio de Janeiro (FAPERJ; process E-26/170.023/2008). The funders had no role in study design, data collection and analysis, decision to publish, or preparation of the manuscript.

**Competing interests:** The authors have declared that no competing interests exist.

## Introduction

The diatom species diversity of the Antarctic is limited by extreme conditions, and studies have led to the conclusion that not all Antarctic islands share a similar flora, although the overall diversity of this flora on the continent is limited compared to tropical and subtropical regions [1]. Furthermore, many of the diatom taxa recently described are endemic to Antarctica and not cosmopolitan [2,3]. Regarding the study of diatom communities, in recent years, a more refined taxonomy has revealed a large number of new species in Antarctica [4–16].

Among biological substrates where microalgal communities may grow, terrestrial macroscopic algae, though unexpected at first, may be included. The class Trebouxiophyceae includes a group of morphologically heterogeneous eukaryotic green algae that occur mainly in soil and continental waters. The genus *Prasiola* Meneghini belongs to this class and includes marine, terrestrial and continental species. *Prasiola crispa* (Lightfoot) Kützing is a terrestrial species of Antarctica, which usually grows on moist soils that are fertilized by bird guano, being more abundant inside and around penguin colonies. These macroalgae tolerate repeated cycles of freezing and thawing during the year, as well as high levels of UV radiation during the summer [17]. At the micro-scale level, the substrate of periphytic communities may provide more than an inert surface, since its physical and chemical characteristics can influence the composition of the community and algal biomass [18–20].

Studies that address distribution patterns often document a decrease in species similarity as spatial distance increases between communities [21–23]. This reduction of similarity is attributed to some major factors such as environmental conditions and species dispersal processes. The influence of spatial distance can be summarized as follows: the larger the distance between sites, the larger are the environmental variations and, therefore, the difference between species tends to be also larger. Spatial distance is a factor that influences the displacement of species across smaller distances, once that this factor increases the chances of survival, and therefore is different from migration which applies over greater distances [21]. The main goal of this study was to evaluate the similarity between diatom communities associated with *P. crispa* spatially distributed in six sites located in the South Shetland Islands, Antarctica. The proposed hypothesis is that the larger the spatial distance between sampling points, the smaller the similarity of diatom communities associated with *P. crispa*.

## Material and methods

During the Brazilian Antarctic Expeditions XXXIII (2014–2015) and XXXIV (2015–2016) (Fig 1), samples of *P. crispa* where obtained from Ardley Island, Halfmoon Island and King George Island (Copacabana, Punta Plaza, Steinhouse and Voureal), in the Maritime Antarctica (Table 1). All collections were authorized by the Secretariat of the Interministerial Commission for Sea Resources (SECIRM) and endorsed by the Brazilian Government's Ministry of the Environment, under the activities regulated by the Brazilian Antarctic Program (PROANTAR).

The samples of *P. crispa* were stored in sealed plastic bags and frozen for further analysis in the Laboratory of Phycology of the Federal University of Santa Maria (UFSM), Rio Grande do Sul, Brazil. To prepare the material, cell contents and organic matter of the samples were removed by oxidation with hydrogen peroxide ($H_2O_2$) and potassium permanganate ($KMnO_4$) [24]. After washing with distilled water, permanent slides were mounted using Naphrax (refractive index of 1.74, Brunel Microscopes Ltd, Chippenham, Wiltshire, United Kingdom), Qualitative and quantitative analyses were carried out under a Leica DM750 optical microscope. The quantitative analyses were performed with approximately 80 valves for each sample, because the abundance is very small in the substrate. Thus, it was necessary to prepare

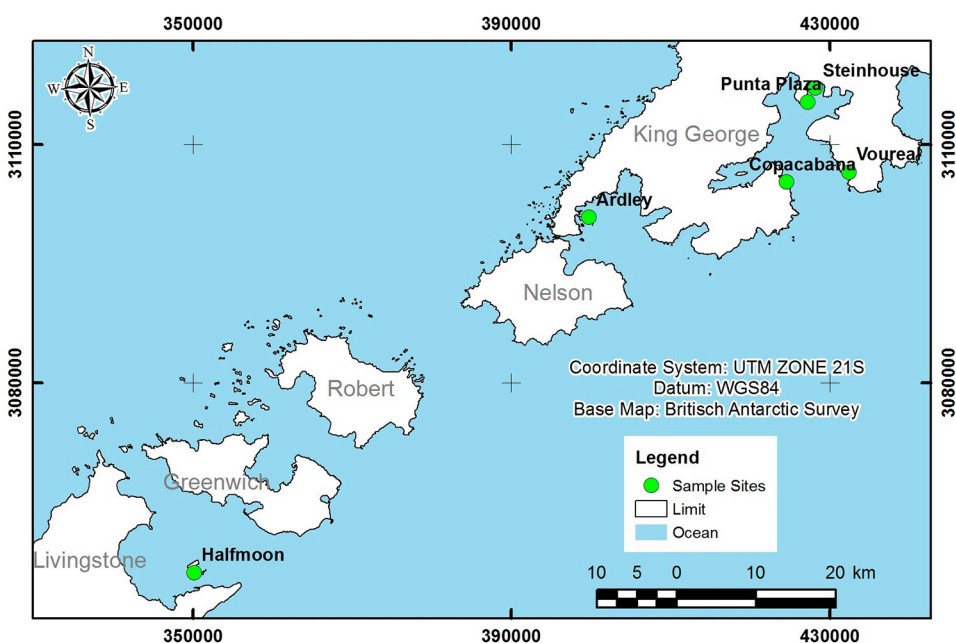

**Fig 1. Map of the location of sampling points where *Prasiola crispa* was collected in Antarctica.** Ardley Island (62˚
12'40.40"S 058˚55'38.70"W), Halfmonn Island (62˚35'41,5"S 059˚55'10,1"W) and King George Island: Copacabana
(62˚10'38,4"S 058˚26'57,3"W), Punta Plaza (62˚5'14.69"S 058˚23'35,21"W), Steinhoose (62˚04'18,9"S 058˚22'28,2"W)
and Voureal (62˚10'2,6"S 058˚17'52.0"W). The Scar Antarctic Digital Database was used as background shapefile to
produce this image and the authors follow the Licence Terms and Conditions under de CC-BY disclamer. More
information can be accessed at https://add.data.bas.ac.uk/repository/entry/show?entryid=f477219b-9121-
44d6-afa6-d8552762dc45.

several permanent slides to make observations under this criterion. Species represented by
only one or two valves in each sample were excluded from the study to avoid potential contam-
inants [25–27]. A specific bibliography for identification of polar diatoms was used [1, 8–11,
14–15, 28–45].

For data analysis, the spatial distance in meters between each pair of sampling sites was cal-
culated based on the registered geographical points. Thus, a geographic distance matrix
(Table 2) was constructed for further comparison using the matrices from biological similarity
of relative abundance (abundance of individuals by taxa) (Table 3) and species composition
(presence and absence of taxa in points).

The estimation of species richness was based on a simple arithmetic of the estimators, with-
out consideration of standard deviations [46]. The samples sufficiency was calculated based on
the percentage of values observed in relation to the estimated values. In order to compare the

**Table 1. Sample site coordinates.**

| Sample Site | Latitude | Longitude |
| --- | --- | --- |
| Ardley Island | 62˚12'40.40"S | 058˚55'38.70"W |
| Halfmoon Island | 62˚35'41.5"S | 059˚55'10.1"W |
| King George Island: Copacabana | 62˚10'38.4"S | 058˚26'57.3"W |
| King George Island: Punta Plaza | 62˚5'14.69"S | 058˚23'35.21"W |
| King George Island: Steinhouse | 62˚04'18.9"S | 058˚22'28.2"W |
| King George Island: Voureal | 62˚10'2.6"S | 058˚17'52.0"W |

**Table 2. Distance table in meters between sampling points where *Prasiola crispa* was collected in Antarctica.**

| Sample sites | Ardley | Copacabana | Halfmoon | Punta Plaza | Steinhouse | Voureal |
|---|---|---|---|---|---|---|
| Ardley | 0 | | | | | |
| Copacabana | 25.17 | 0 | | | | |
| Halfmoon | 66.76 | 89.14 | 0 | | | |
| Punta Plaza | 31.07 | 10.43 | 97.19 | 0 | | |
| Steinhouse | 32.74 | 12.37 | 99.00 | 1.98 | 0 | |
| Voureal | 33.13 | 7.96 | 96.46 | 10.20 | 10.55 | 0 |

species richness among the sample points, individual rarefaction curves were constructed, and compared in the lowest abundance value among them (N = 41), in order to correct for differences in size and sample effort. For this purpose, the abundance data of each species were used for each sampling point. The EstimateS program (Department of Ecology & Evolutionary Biology, University of Connecticut, Storrs, USA) was used for analyses, with 100 randomisations of the data [47]. Rank-abundance diagrams were constructed to describe and compare the distribution of abundance and dominance of species at each sampling site using relative abundance data [48]. To describe these distributions numerically, the Pielou equitability index was adopted, based on the Shannon diversity index.

**Table 3. List of all observed species with their acronyms and abundance in the samples from Antarctic Islands.**

| Species | Acro | Sample points | | | | | |
|---|---|---|---|---|---|---|---|
| | | Ard | Cop | Hal | Pup | Ste | Vou |
| *Luticola amoena* Van der Vijver, Kopalová, Zidarova & Levkov | A | 0 | 0 | 0 | 0 | 5.57 | 0 |
| *Luticola austroatlantica* Van de Vijver, Kopalová, S.A. Spaulding & Esposito | B | 0 | 0 | 0 | 0 | 2.29 | 0 |
| *Luticola muticopsis* (Van Heurck) D.G. Mann | C | 0 | 0 | 0 | 0 | 1.97 | 1.35 |
| *Achnanthes* sp. | D | 0 | 0 | 0 | 2.41 | 0 | 0 |
| *Achnanthidium* aff *indistinctum* Van de Vijver & Kopalová | E | 0 | 1.31 | 0 | 0 | 10.49 | 0 |
| *Cyclotella meneghiniana* Kützing | F | 0 | 1.74 | 0 | 0 | 0 | 0 |
| *Cocconeis pinnata* var. *matsii* Al-Handal, Riaux-Gobin & Wulff | G | 2.44 | 44.12 | 10.59 | 12.65 | 7.22 | 11.97 |
| *Eunotia* aff *pseudopaludosa* Van de Vijver, de Haan & Lange-Bertalot | H | 0 | 0 | 32.71 | 21.68 | 24.92 | 2.51 |
| *Fragilaria* cf *parva* Tuji & D.M. Williams | I | 2.44 | 4.36 | 1.87 | 3.32 | 0.33 | 0 |
| *Hantzschia amphioxys* f. *muelleri* Ts. KoBayashi | J | 7.32 | 4.36 | 0 | 0 | 0 | 0 |
| *Pteroncola carlinii* Almandoz & Ferrario | K | 0 | 0 | 0 | 8.43 | 7.54 | 0 |
| *Mayamaea* cf *atomus* (Kützing) Lange-Bertalot | L | 0 | 0 | 4.98 | 0 | 0 | 0 |
| *Navicula* aff *perminuta* Østrup | M | 0 | 2.18 | 0 | 0 | 0 | 0 |
| *Pinnularia* aff *microstauron* (Ehrenberg) Cleve | N | 0 | 3.06 | 3.43 | 0 | 0 | 0 |
| *Pinnularia australoschoenfelderi* Zidarova, Kopalová & Van de Vijver | O | 51.21 | 0 | 13.39 | 0 | 0 | 53.86 |
| *Pinnularia austroshetlandica* (Carlson) Cleve-Euler | P | 24.39 | 9.61 | 0 | 4.82 | 6.23 | 17.57 |
| *Pinnularia borealis* Ehrenberg sensu lato | Q | 12.19 | 0 | 3.12 | 9.94 | 14.75 | 0 |
| *Psammonthidium rostrogermainii* Van de Vijver, Kopalová & Zidarova | R | 0 | 2.18 | 0 | 0 | 6.56 | 0 |
| *Psammothidium germainii* (Manguin) Sabbe | S | 0 | 0 | 7.78 | 0 | 3.94 | 0 |
| *Psammothidium papilio* (D.E. Kellogg et al.) Kopalová & Zidarova | T | 0 | 13.1 | 2.18 | 7.23 | 8.19 | 9.85 |
| *Pseudogomphonema kamtschaticum* (Grunow) L.K.Medlin | U | 0 | 6.98 | 4.05 | 0 | 0 | 2.89 |
| *Luticola olegsakharovii* Zidarova, Levkov & Van de Vijver | V | 0 | 0 | 8.09 | 20.18 | 0 | 0 |
| *Thalassiosira gracilis* var.*expecta* G. Fryxell & Hasle | X | 0 | 7.86 | 7.78 | 0 | 0 | 0 |

Acro = Acronyms; Ard = Ardley; Cop = Copacabana; Hal = Halfmoon; PuP = Punta Plaza; Ste = Steinhouse; Vou = Voureal.

The diversity was partitioned into the alpha, beta and gamma components using Hill's numbers [49,50] with the "entropart" package [51] in the R program (R Foundation for Statistical Computing, Vienna, Austria) (R Core Team 2017), with the approach of multiplicative partitioning of diversity.

Jaccard and Bray-Curtis dissimilarity coefficients were used to calculate dissimilarity between pairs of sampling points based on incidence and abundance data. An array of geographical (Euclidean) distances between points was derived from the longitude and latitude coordinates. Subsequently, the Jaccard and Bray-Curtis distance matrices were regressed with the geographic distance matrix using linear regressions [52].

The standard coefficients of these regression models were used as measures of the decay rate when comparing similarity as a function of geographical distance between points [22]. In addition, biotic distance data were correlated with the geographic distance matrix in order to corroborate the relationship probability values based on 10,000 permutations using the Mantel test [23]. Statistical analyses were performed in the R program using the "vegan" packages [53] and "betapart" [54] packages, and visualised through dendrograms also using the "vegan" package [53] in R environment.

In order to verify which component of the beta diversity predominantly contributed to the dissimilarity between sampling points, species presence/absence dissimilarity was decomposed into substitution (turnover) and nestedness [54] components, while abundance based dissimilarity was decomposed into the components of abundance balanced variation and abundance gradients [55]. For this, Jaccard and Bray-Curtis dissimilarity coefficients were used in the "betapart" package [56] implemented in the R program. The matrices of beta diversity components based on presence / absence and abundance were also associated to the geographic distance matrix, as previously described, in order to verify which dissimilarity component was related to the spatial configuration of the sampling sites.

The protocols used for the present study can be accessed through protocols (dx.doi.org/10.17504/protocols.io.6tcheiw).

## Results and discussion

The diatom flora associated with *Prasiola crispa* at the six sampling sites were represented by 23 species (Figs 2 and 3) distributed in 15 genera. Two species are typical of marine environments: *Pseudogomphonema kamtschaticum*, *Cocconeis pinnata* var. *matsii* and *Pteroncola carlinii*. The presence of these species in these terrestrial sites is probably related to their proximity to the sea.

Comparing the rarefied richness of the islands, in order to correct for sample size differences between sites, we observed that Steinhoose and Halfmoon had higher species richness, even when comparing the same number of individuals (total abundance = 41 in both sites) (Fig 4). Intermediate richness was observed in Copacabana and Punta Plaza, and even though Voureal had the largest number of individuals, this site had similar rarefied richness to that of Ardley, which had the lowest number of species and individuals (Fig 4).

The most abundant species were *Pinnularia australoschoenfelderi* (20.0%), *Luticola austroatlantica* (15.5%), *Luticola amoena* (15.3%), *Pinnularia austroshetlandica* (9.2%) and *Psammothidium papilio* (8.0%), which altogether represented 68% of the total number of individuals collected. There was a difference in the most abundant species between each sampling site (Fig 5). The dominance pattern in Steinhoose and Punta Plaza, the two areas closest to each other in this study, was similar, with high abundance of *L. austroatlantica* and *P. borealis*. These two areas also had the highest Pielou equitability values (Punta Plaza: Pielou = 0.91, Steinhoose: Pielou = 0.89). However, *L. austroatlantica*, was also the most abundant at the

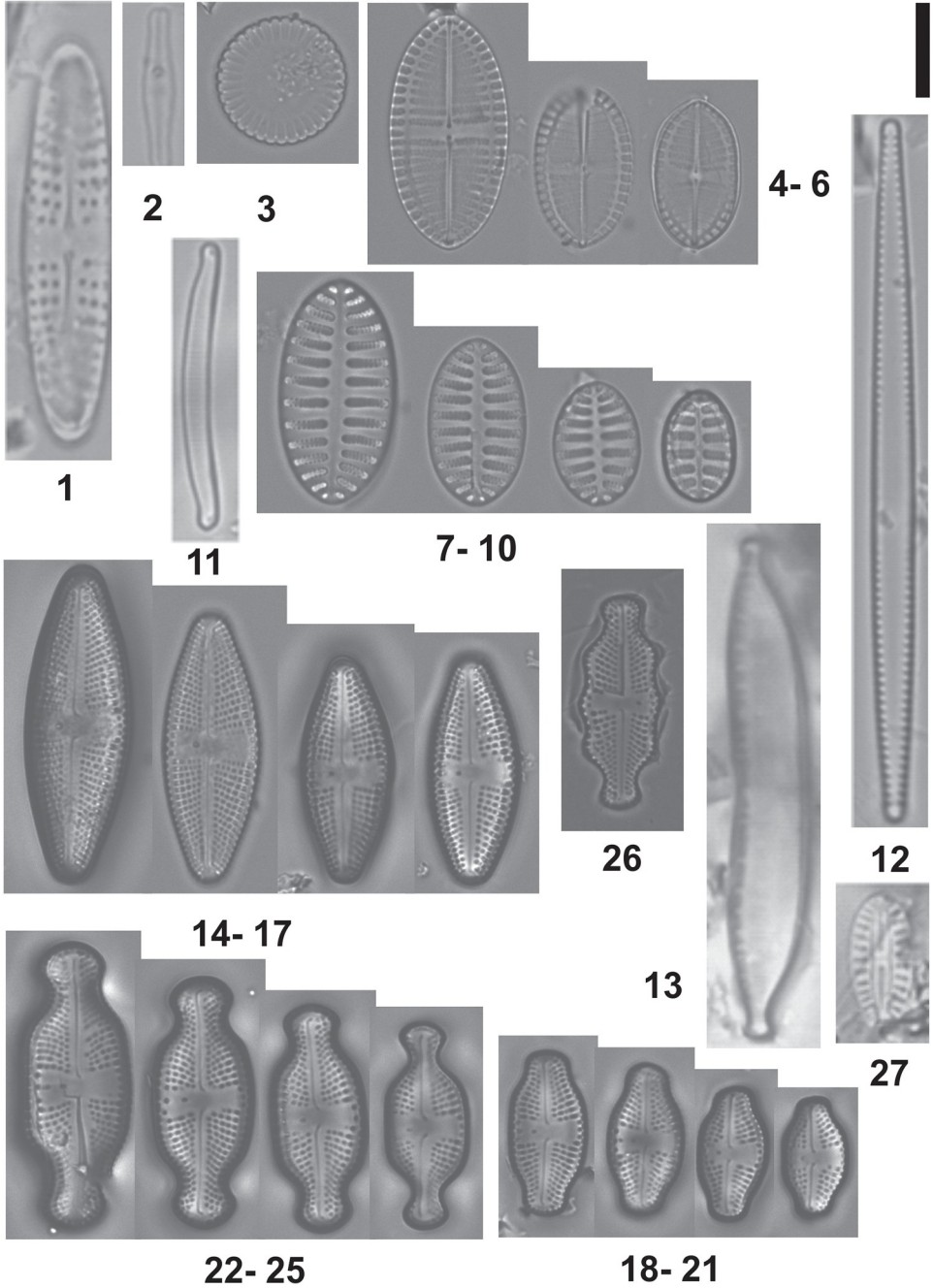

**Fig 2. Taxa of diatoms in LM.** 1. *Achnanthes* sp, 2. *Achnanthidium aff indistinctum*, 3. *Cyclotella meneghiniana*, 4–10. *Cocconeis pinnata* var. *matsii*, 11. *Eunotia aff pseudopaludosa*, 12. *Fragilaria* cf *parva*, 13. *Hantzschia amphioxys*, 14–17. *Luticola amoena*, 18–21. *Luticola austroatlantica*, 22–25. *Luticola muticopsis*, 26. *Luticola olegsakharovii* e 27. *Mayamaea* cf *atomus*. LM scale bar = 10 μm.

other extreme of the distance gradient in Halfmoon (Pielou = 0.86). *P. australoschoenfelderi* dominated in Ardley (Pielou = 0.73) and Voureal (Pielou = 0.71), while *L. amoena* dominated in Copacabana (Pielou = 0.76). This species was also among the three most abundant species in Halfmoon, Punta Plaza and Voureal.

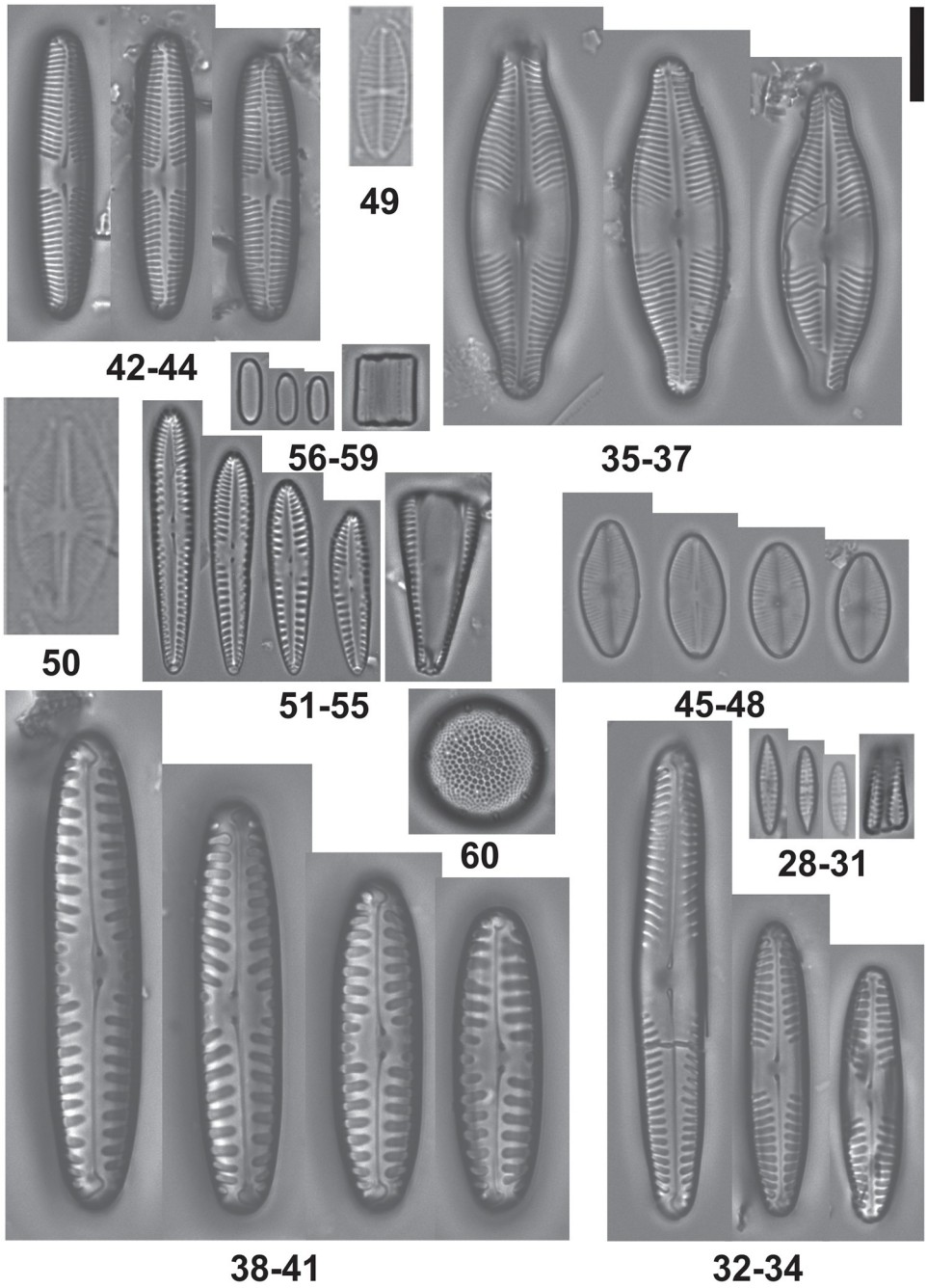

**Fig 3. Taxa of diatoms in LM.** 28–31. *Navicula aff perminuta*, 32–34. *Pinnularia australoschoenfelderi*, 35–37. *Pinnularia austroshetlandica*, 38–41. *Pinnularia borealis*, 42–44. *Pinnularia aff microstauron*, 45–48. *Psammothidium germainii*, 49. *Psammothidium papilio*, 50. *Psammonthidium rostrogermainii*, 51–55. *Pseudogomphonema kamtschaticum*, 56–59. *Pteroncola carlinii* e 60. *Thalassiosira gracilis var.expecta*. LM scale bar = 10 µm.

The diversity partition demonstrated that true diversity based on species richness (greater weight on rare species), Shannon's entropy (similar weight on common and rare species), and Simpson index (greater weight on dominant species) showed similar response patterns, with higher values in Steinhoose (Table 4). The alpha diversity was approximately ten species for

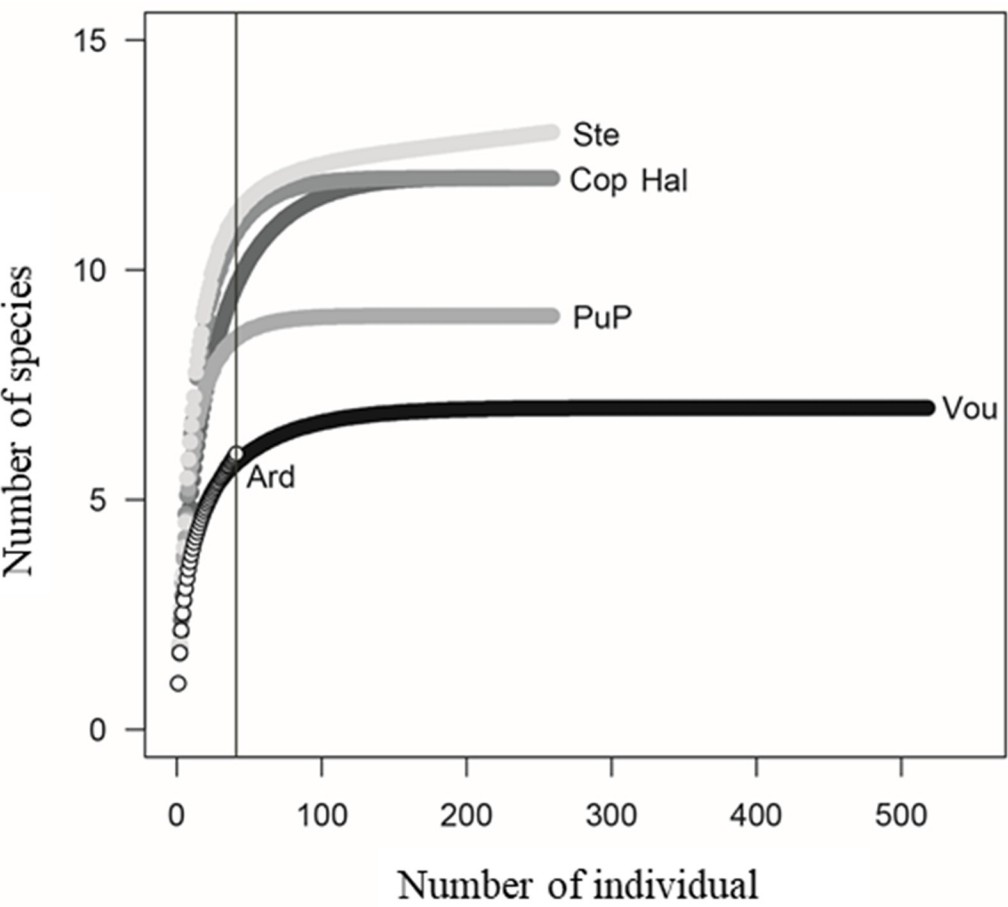

**Fig 4. Rarefaction curves of the diatom species associated with *Prasiola crispa* for the sampled sites.** The vertical line represents the point of comparison between the islands. Confidence intervals were omitted for better visualization. Ard = Ardley; Cop = Copacabana; Hal = Half Moon; PuP = Punta Plaza; Ste = Steinhoose; Vou = Voureal.

species richness; this analysis further showed that the true beta diversity of Shannon index ($^1D$) was lower than the richness based on species richness ($^0D$) and based on the inverse concentration of Simpson index ($^2D$). This shows that differences between sampling sites occurred due to the contrast between the most abundant and the rarest species at each site (Table 4). The beta diversity values for all orders were close to 2, indicating that there are at least two large groups of species with high substitutions.

Total dissimilarity based on the presence / absence and abundance of species between the islands was 80.7% and 80.5%, respectively. Similarly, analyses of parity-by-pair dissimilarity, both presence / absence and abundance based, showed similar patterns of response, but no formation of distinct groups (Fig 6). The high similarity between the nearest sites, Steinhoose and Punta Plaza, with the most distant site, Halfmoon, for presence / absence as well as for number of individuals, should be emphasized. Copacabana had the highest average dissimilarity with the other sampling sites based on species composition (Fig 6A), while Ardley had the greatest dissimilarity based on abundance of individuals (Fig 6B).

The decomposition of beta diversity based on the presence / absence of species showed that the species substitution component was the most important relative to dissimilarity among the sampling sites (73.1%). The relative importance of the nesting component was only 7.6%,

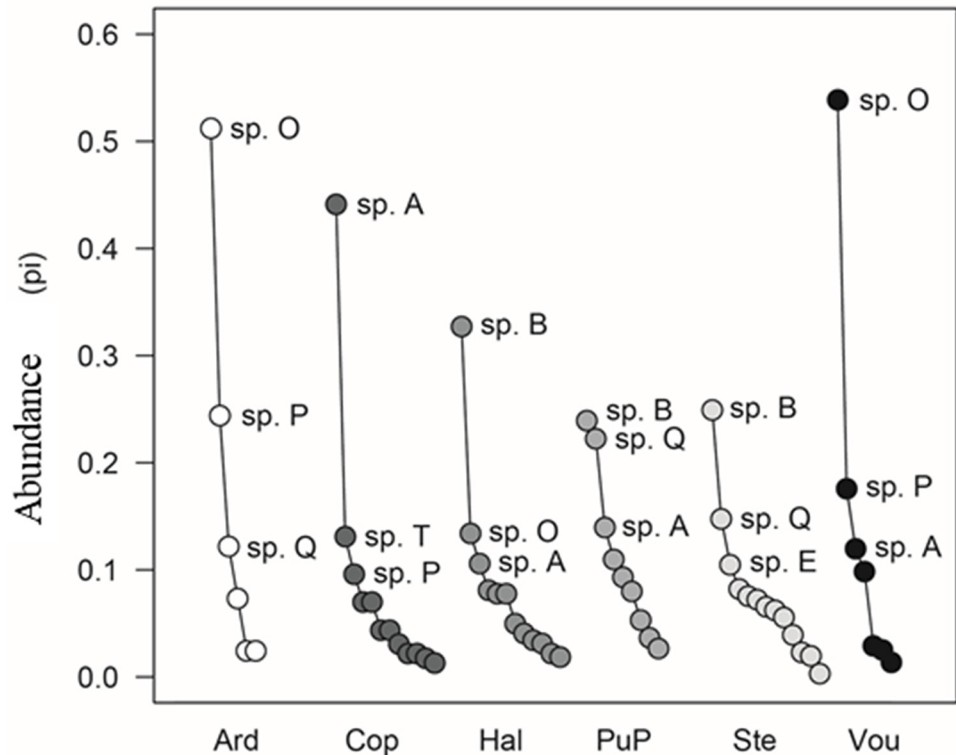

**Fig 5. Plots of abundance (proportional [pi]) of the diatom species associated with *Prasiola crispa* for the sampled sites.** The abbreviations refer to the names of the species in Table 2.

corroborating the beta diversity values slightly higher than two shown by the partitioning analysis of diversity in the alpha, beta and gamma components. These results were strongly influenced by the observed species richness in each sampling site, as well as the high number of exclusive species (26.1% of the total species). In addition, only 26.1% of the species occurred in more than 50% of the sites.

Community similarity between sampling sites was not related to the geographical distance between them, based on incidence (t = -0.509, p = 0.619) and abundance (t = -1.041, p = 0.3167) (Fig 7A and 7D). Species substitution components (t = 0.610, p = 0.553) and nesting (t = -0.403, p = 0.693) of incidence-based beta diversity were also not related to geographical distance (Fig 7B and 7C). Similarly, beta diversity components based on the abundance from balanced variation of abundance (t = 0.180, p = 0.860) and abundance gradients

**Table 4. Multiplicative partitioning of the diatom species diversity associated with *Prasiola crispa* for the sampled sites.**

| | Islands | | | | | | Components of diversity | | |
|---|---|---|---|---|---|---|---|---|---|
| | **Ard** | **Half** | **Cop** | **PuPl** | **Stei** | **Vou** | **A** | **β** | **γ** |
| $^0D$ | 6 | 12 | 12 | 9 | 13 | 7 | 10 | 2.3 | 23 |
| $^1D$ | 3.73 | 8.45 | 6.68 | 7.31 | 9.75 | 3.95 | 6.38 | 1.92 | 12.26 |
| $^2D$ | 2.91 | 6.17 | 4.22 | 6.32 | 7.92 | 2.88 | 4.44 | 2.01 | 8.92 |

Ard = Ardley; Half = Halfmoon; Cop = Copacabana; Pupl = Punta Plaza; Stei = Steinhouse; Vou = Voureal; true beta diversity of Shannon ($^1D$); species richness ($^0D$); concentration of Simpson ($^2D$).

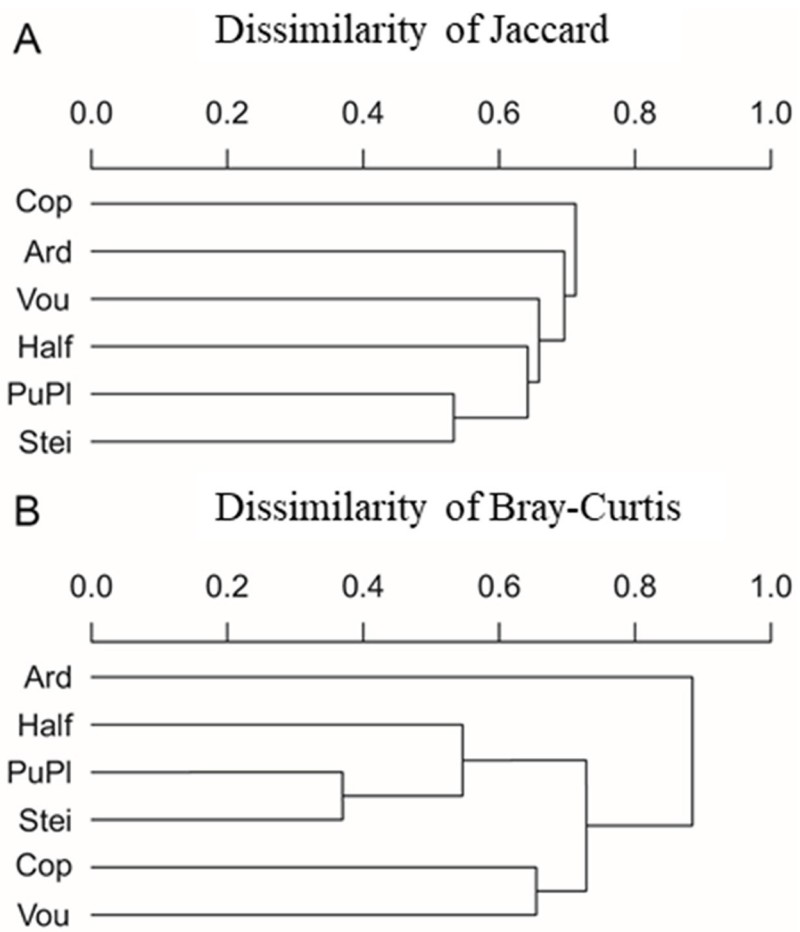

**Fig 6. Average dissimilarity between sampling points of the diatom species associated with *Prasiola crispa* for the sampled sites based on the presence / absence of species (A) and abundance (B).**

(t = 0.654, p = 0.525) were not related to the geographical distance between sampling sites (Fig 7E and 7F). The Mantel tests, based on 10.000 permutations, corroborated the probability values for this lack of relationship between the similarity matrix and dissimilarity components with the between-sites geographical distance. The lack of a positive or negative similarity relationship with geographical distance occurred due to the high similarity in composition and abundance among the most distant sampling points. In other words, even very close or very distant sites have high similarity in diatom composition and abundance, which resulted in the absence of a linear relationship pattern withdistance. It is possible, however, that if the distances between samples were greater, a clearer relationship might have been observed. According to [57], when studying phytoplankton, at small scales in general there is no effect of the distance, and environmental conditions are more significant. The composition of macroscopic organisms in a habitat seems to be more affected by geographical distances than microorganisms, which is mainly shaped by local conditions [58]. On the other hand, [59], in diatom communities of microbial mats of Antarctic ponds within a similar distance range as our study, found considerable spatial variation that could not be explained by local physical and chemical variables. They concluded that the history of dispersal and colonization of diatoms played an important role in their community structures.

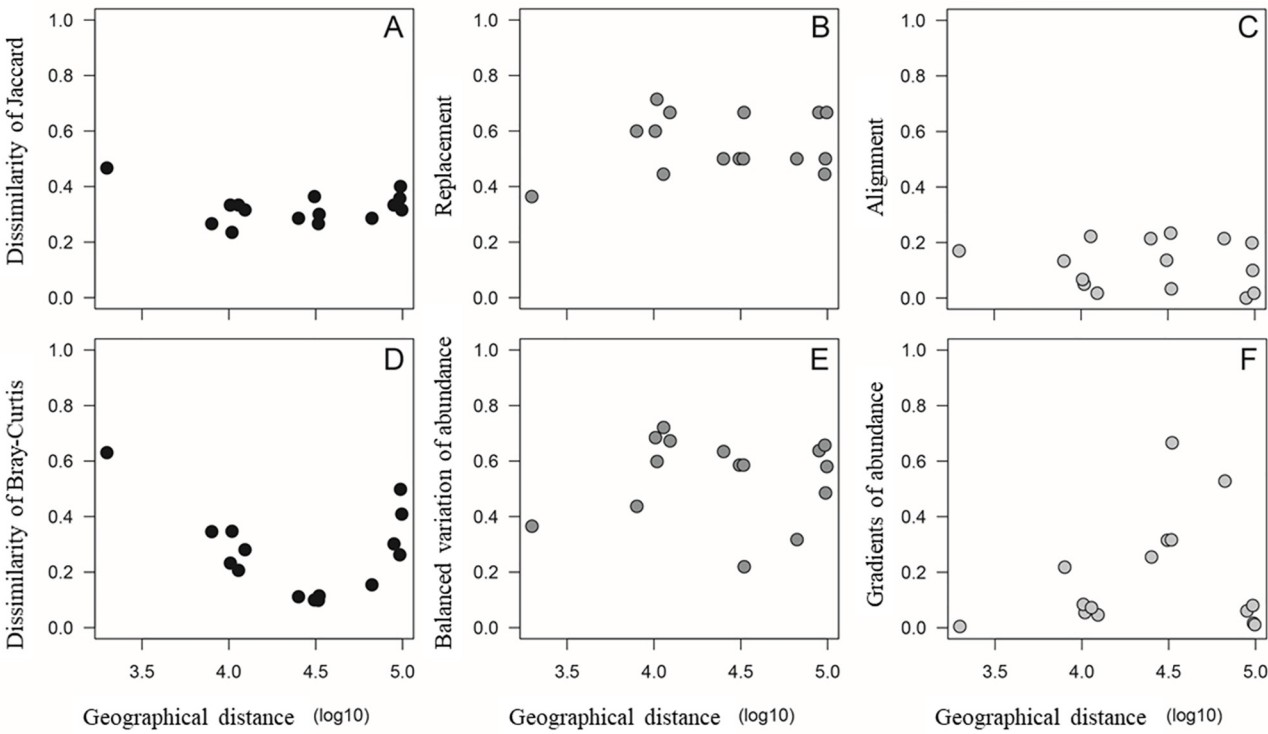

**Fig 7. Similarity of Jaccard (A) decomposed into components of substitution (B) and nesting (C), and similarity of Bray-Curtis (D) decomposed into components of balanced variation of abundance (E) and abundance gradients (F) of the diatom community associated with *Prasiola crispa* related to the geographical distance between the sampling sites.**

*Prasiola crispa* is found in drainage lines in Antarctica and because it is ornithocoprophilous, presents considerable biomass mainly around penguin colonies [60], which makes nutrient availability for both the green algae and the surrounding diatom community higher, when compared to e.g. epilithic habitats. The lack of differentiation between the communities with geographic distance could also be attributed to the lower climatic variation in Antarctica compared to tropical and subtropical regions. In a previous study on three marine red macroalgae in Newfoundland Bay, Antarctica [61], the biophysical-chemical characteristics of each sampling site affected the diatom communities more markedly than the different species of host macroalgae or even depth of sampling. Moreover, the diversity of the epiphytic community of diatoms associated with red macroalgae collected near the sea ice of Cape Evans, Antarctica, showed that species diversity decreased as the depth below the sea ice increased, whereby the dominant taxa had also changed in relation to depth [62, 63].

Our study showed that even with marked distance between sampling points, there was high similarity in the composition and abundance of the diatom communities, which resulted in the absence of a linear relationship pattern with distance. However, it was possible to demonstrate that regarding dominant species, the closest sampling sites were more similar. Thus, the inexpressive differentiation of the communities showed that *P. crispa* as a substrate seems to be an important factor for the selection of the existing epiphytic community.

There have been an increasing number of studies reporting changes in the community of Antarctic organisms due to global warming. Some penguin communities are experiencing population declines as a result of rising temperatures [64] and moss communities are increasing on King George Island, colonizing fields in uncovered areas due to shrinking glaciers [65].

Decreases in microbial diversity patterns has also been reported in regions of Patagonian and other Antarctic lakes, mainly found to decrease in diatom diversity [66,67], as well with evidence for the decline of phytoplankton [68] and bacterioplankton [69]. Such aspects lead us to reflect on how much the diatom communities as well as the substrate of the present study, *P. cripa* can be affected since it grows in humid places with high nutrient input. If we eleminate these factors, we would also affect the existence of this macroalgae species and, consequently, that of the associated aerophytic diatom community. Although we have not observed any exclusive communities present in this substrate we believe that by affecting the stability of *P. crispa* in terrestrial wetlands we would also affect the dispersal of associated diatoms.

## Acknowledgments

We acknowledge the Coordination of Improvement of Higher-Level Personnel (CAPES), for the scholarship granted to the first author. The authors would also like to thank the brazilian Ministry of the Environment and the Inter-Ministerial Secretary for Sea Resources (SECIRM) for the autorizations granted for field collection activities under the Brazilian Antarctic Program. We would like to thank the researchers at Luxembourg Institute of Science and Technology, Luc Ector and Carlos Wetzel, for helping to species identification.

## Author Contributions

**Conceptualization:** Juliana Ferreira da Silva, Maria Angélica Oliveira Linton, Filipe de Carvalho Victoria.

**Data curation:** Juliana Ferreira da Silva, Maria Angélica Oliveira Linton.

**Formal analysis:** Juliana Ferreira da Silva, Filipe de Carvalho Victoria.

**Investigation:** Juliana Ferreira da Silva, Maria Angélica Oliveira Linton, Eduardo Pereira da Silva, Rodrigo Paidano Alves, Filipe de Carvalho Victoria, Margéli Pereira de Albuquerque.

**Methodology:** Juliana Ferreira da Silva, Raylane Ribeiro da Anunciação, Eduardo Pereira da Silva, Rodrigo Paidano Alves, Adriano Luis Schünemann, Margéli Pereira de Albuquerque.

**Supervision:** Maria Angélica Oliveira Linton, Antônio Batista Pereira.

**Validation:** Juliana Ferreira da Silva, Adriano Luis Schünemann.

**Visualization:** Adriano Luis Schünemann.

**Writing – original draft:** Juliana Ferreira da Silva, Maria Angélica Oliveira Linton.

**Writing – review & editing:** Juliana Ferreira da Silva, Maria Angélica Oliveira Linton, Adriano Luis Schünemann, Filipe de Carvalho Victoria, Margéli Pereira de Albuquerque, Antônio Batista Pereira.

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
