## [Decision Letter · Decision Letter 0]

1 Aug 2019

PONE-D-19-16073

Distribution of diatom communities associated with terrestrial green macroalgae in the South Shetland Islands, Maritime Antarctica

PLOS ONE

Dear Dr. Victoria,

Thank you for submitting your manuscript to PLOS ONE. After careful consideration, we feel that it has merit but does not fully meet PLOS ONE’s publication criteria as it currently stands. Therefore, we invite you to submit a revised version of the manuscript that addresses the points raised during the review process.

We would appreciate receiving your revised manuscript by Sep 15 2019 11:59PM. To enhance the reproducibility of your results, we recommend that if applicable you deposit your laboratory protocols in protocols.io, where a protocol can be assigned its own identifier (DOI) such that it can be cited independently in the future. For instructions see: http://journals.plos.org/plosone/s/submission-guidelines#loc-laboratory-protocols

We look forward to receiving your revised manuscript.

Kind regards,

Jinzhuang Xue

Academic Editor

PLOS ONE

Journal Requirements:

1. Our internal editors have looked over your manuscript and determined that it may be within the scope of our Life in Extreme Environments Call for Papers. The Collection will encompass a diverse range of research articles to better understand life and biogeochemistry in extreme environments. Additional information can be found on our announcement page: https://collections.plos.org/s/extreme-environments. If you would like your manuscript to be considered for this collection, please let us know in your cover letter and we will ensure that your paper is treated as if you were responding to this call. If you would prefer to remove your manuscript from collection consideration, please specify this in the cover letter.

3. In your Methods section, please provide additional location information, including geographic coordinates for the data set if available.

4. We note that [Figure(s) 1] in your submission contain [map/satellite] images which may be copyrighted. All PLOS content is published under the Creative Commons Attribution License (CC BY 4.0), which means that the manuscript, images, and Supporting Information files will be freely available online, and any third party is permitted to access, download, copy, distribute, and use these materials in any way, even commercially, with proper attribution. For these reasons, we cannot publish previously copyrighted maps or satellite images created using proprietary data, such as Google software (Google Maps, Street View, and Earth). For more information, see our copyright guidelines: http://journals.plos.org/plosone/s/licenses-and-copyright.

1.    You may seek permission from the original copyright holder of Figure(s) [1] to publish the content specifically under the CC BY 4.0 license. 

5. Thank you for including your competing interests statement; "NO"

6. Thank you for including your funding statement; "No"

Please provide an amended Funding Statement that declares *all* the funding or sources of support received during this specific study (whether external or internal to your organization) as detailed online in our guide for authors at http://journals.plos.org/plosone/s/submit-now.  

Please state what role the funders took in the study.  If any authors received a salary from any of your funders, please state which authors and which funder. If the funders had no role, please state: "The funders had no role in study design, data collection and analysis, decision to publish, or preparation of the manuscript."

7. Please include your tables as part of your main manuscript and remove the individual files. Please note that supplementary tables (should remain/ be uploaded) as separate "supporting information" files

Reviewers' comments:

Reviewer's Responses to Questions

**Comments to the Author**

1. Is the manuscript technically sound, and do the data support the conclusions?

Reviewer #1: Yes

Reviewer #2: Yes

2. Has the statistical analysis been performed appropriately and rigorously? 

Reviewer #1: Yes

Reviewer #2: Yes

3. Have the authors made all data underlying the findings in their manuscript fully available?

Reviewer #1: Yes

Reviewer #2: Yes

4. Is the manuscript presented in an intelligible fashion and written in standard English?

Reviewer #1: No

Reviewer #2: Yes

5. Review Comments to the Author

Reviewer #1: Distribution of diatom communities associated with terrestrial green macroalgae in the South Shetland Islands, Maritime Antarctica

Juliana Ferreira da Silva et al.

This study reports diatom diversity in communities associated with the alga Prasiola crispa in terrestrial habitats in the South Shetland Islands, maritime Antarctic, as well as attempting to examine whether diversity of specific communities was related in any way to separation distance from other locations examined. In reality most locations were on King George Island, while Ardley Island is only separated from that by a tidally inundated isthmus. As such, it is an original study, contributing to the knowledge of this group in Antarctica.

Language is generally clear, though some minor syntax and spelling editing is required throughout; I note that this journal does not offer copy editing, so this will have to be done carefully during revision. Is the citation format in the text correct for the journal? It is unusual to list both numbered references and fully named author lists together.

Abstract – the description of the role of distance (‘isolation by distance’) in structuring (or not) diatom community composition could be reworded a bit to make it clearer: at present the sentence structure seems to suggest initially that distance is being considered from each of the six sampling sites individually (for instance along a transect from a central sampling location at each), and rather it later becomes clear that the intention is to look at distance as a factor between the six sampling locations.

Introduction – the opening para should make it clear from the outset that the focus is on diatoms; at present the opening sentences appear to make statements applying to the Antarctic flora more generally – if this is the case then they need expanding into a separate general opening para, before subsequently focussing in on diatoms specifically. However, I think the intention is to focus on diatoms from the outset.

Terminology in the para describing the influence of distance – it should be ‘community similarity’ rather than ‘species similarity’.

Within the methods section there is no reference given to how the diatoms obtained were identified? I appreciate the long list of recent species descriptions early in the Introduction, but something specific should be said in this section. Other than these comments, the approaches and analyses used are appropriate.

Taxonomic authorities – I believe that journal format only requires these to be listed once, so there are many instances where repeat listings can be deleted. If acceptable to the journal, I would suggest placing all authorities within Table 2 and deleting them from the text, which will also help make the text more readable.

Diversity indices – I am not familiar with what 0D, 1D, 2D annotation means?

Overall, while the study found differences in various indices between sites, community similarity was high across the entire study area, emphasised by the highest similarity being found between the most distant sites, and there was no overall relationship between separation distance and diversity. I wonder if this might reflect the overall physical scale of the study area, where most of the sampling sites were actually quite close to each other – in other words does this indicate the scale of the study was not large enough to identify such effects, and that ‘isolation’ was not a significant factor at this scale? Perhaps a larger geographical sampling scale would lead to identification of such a relationship, as predicted by the study’s overall hypothesis?

Figures – the labels on the diatom photo figs are blurred in my pdf?

Table 1 – I think I would suggest presenting this data in km rather than m

Peter Convey

British Antarctic Survey

19 July 2019

Reviewer #2: Review for PLOS one of manuscript PONE-D-19-16073 by da Silva et al.

In this manuscript the authors analyze a set of epiphytic diatom samples collected from the terrestrial macroalgae Prasiola crispa from six sites located in the South Shetland Islands, Maritime Antarctica. They observed rather species-poor and homogeneous diatom communities. In addition to reporting the species list, they used various statistical methods on their abundance data to demonstrate that the geographical distance is not the factor driving the distribution of the diatoms identified in their samples.

In my opinion this is a valuable diatom dataset collected from an interesting habitat in a very remote region. The authors provide high-quality illustrations of the diatom taxa they could observed (obtained with both light and scanning electron microscopes). Their conclusions are supported by the statistical analyses carried out. I think this study is of interest for researchers interested in the distribution of these microorganisms in such extreme habitat. I think this manuscript requires some corrections and improvements before it can be published in PLoS One. Please see below my main suggestions, followed by a list of specific comments.

General comments:

Diatom identification: In the method section of the manuscript, the authors only referenced Round et al. (1990) as the source for the identification of the diatoms they observed in their samples. This is not suitable as this book by Round et al. only deals with taxonomy at the generic level and some of the genera the authors identified where not even described at the time it was published (e.g. Psammothidium, Mayamaea). It would be much better to list all the papers and flora used for identification. In addition, I found particularly surprising that you did not make use of the flora published by Zidarova et al. (2016) specifically dealing with Maritime Antarctica, including the South Shetlands Islands. In particular, the species the authors identified as Luticola aff beyensis and Luticola muticopsis morphotype 1 may be Luticola amoena and Luticola austroatlantica, respectively, as illustrated in the book by Zidarova et al. (2016). Cocconeis aff costata may also correspond to Cocconeis pinnata var. matsii as described by Al-Handal et al. (2010) from King George Island.

Spelling of diatom names: I’ve found numerous mistakes throughout the text of the manuscript, the legend of the figures and in the tables. Please pay attention to that and correct them.

Rarefaction analysis: Taxonomic richness assessed through rarefaction in fact consists in an interpolation to lower (and common to all sample) count size and it results in a loss of information (Giesecke et al. 2014). Moreover, samples standardized by size can have different degrees of completeness, depending on the species-abundance distributions of the assemblages that are compared. A better way is to compare samples of equal completeness, not equal size (Chao and Jost 2012). This can be done by using the species accumulation curve to extrapolate the total species richness of the less complete samples (Béguinot 2015). Several parameters can be used to estimate the completeness of the sampling such as the percentages of species represented by one individual (=singleton) and the sample intensity, i.e. abundance divided by richness (Lopez et al. 2012).

Comparison of the results: the main result is that spatial distance does not appear to drive differentiation in diatom communities in this study. Maybe this result is due to the fact that the distance covered in the study is relatively small (less than 100 km). Could you compare your result with similar studies dealing with the effect of spatial distance on diatom community composition?

Conclusion: I think the manuscript would really gain if you could broaden a little the scope of your conclusions with a few sentences. For example, could you discuss about any wider implications regarding the potential effect of further climate warming on the resilience of this community. I read that the penguin populations were threatened as their conditions for nesting are changing (change in local weather with unprecedented rain, prematurely snow melt creating puddles of water on the ground, etc…). How would this affect Prasiola and its associated diatom flora?

Specific comments:

Page 1: In the title, I would use: Distribution of aerophilous diatom communities … to make it clear what kind of samples were investigated.

Page 1: in the address of the authors, correct spelling of Antarctic (only one “t”)

Page 2, in abstract: rephrase such as: …such as the type of substrate…

Page 2: correct spelling of Luticola muticopsis (not multicopsis)

Page 2: add dots after initials such as: D.E. Kellogg et al.

Page 2:… in our samples, …

Page 2: …and abundance of their diatom community…

Page 2: keywords: replace “Diatom” by “aerophilous diatoms”, delete “Taxonomy” as this paper does not deal with taxonomy

Page 3: rewrite the sentence such as: The larger the distance between sites, the larger are the environmental variations and, therefore, the difference in species composition tends to be also larger. The following sentence, starting by “Spatial distance is a factor that influences…” is unclear, please re-write it.

Page 4: quantitative analyses: give the number of valves counted to obtain the relative abundance data

Page 4: the statement starting with “sample sufficiency…” is unclear

Page 4: regarding rarefaction, please see above in general comments

Page 5: separate thousands using a comma such as: 10,000

Page 5: use lower case for “also”

Page 5: …(Oksanen et al. 2016) in R.

Page 5: correct spelling of Pseudogomphonema and carlinii

Page 5: Cocconeis pinnata var. matsii is also a marine species (if it’s the correct identification?)

Page 6: correct spelling for muticopsis (several times)

Page 6: just give the authorities once, when a species is first mentioned. There is no need to repeat them every time thereafter.

Page 7: separate thousands using a comma such as: 10,000

Page 9: delete “1.” before “Hamsher”

Page 9: just use Fottea for title of the journal, delete “Czech Phycological Society – Praha, Czech Republic, 2007, currens.”

Page 9: in the reference by Kochman-Kedziora et al.: just use Fottea for title of the journal, delete “Czech Phycological Society – Praha, Czech Republic, 2007, currens.”

Page 10: delete “3.” before “Kopalova”

Page 11: use capital L for journal title: Limnetica

Page 11: delete “2.” before “Van de Vijver”

Fig. 1: add an insert showing the whole of Antarctica to help the reader immediately visualize the location of this region. Some locations on the map and the legend are in Portuguese (e.g Rei George instead of King George; Sistema de Coordenadas, etc…). Please change all to English. Also correct spelling of British. Also give explanation about the units used on sides of the figure.

Figs 2 and 3: indicate to what distance the scale bar corresponds (is it 5 microns?), either directly on the figures or in the legend. In the legend: add “LM photographs”; correct spelling of diatom names, use English instead of Portuguese for morphotype

Fig. 4: add “SEM photographs”, correct spelling of diatom names.

Table 1: I would give the distances in kilometers instead of meters. If you insist in using meters, at least delete the decimals…

Table 2: There are many mistakes in the diatom names:

Muticopsis instead of multicopsis

Cyclotella instead of Ciclotella

Fragilaria instead of Fragillaria

Hantzschia instead of Hantzchia

Carlinii instead of carlini

Pseudogomphonema instead of Pseudoghophonema

Olegsakharovii instead of olegsakharoni

References used in this review:

Al-Handal, A., Riaux-Gobin, C., Wiulff, A. (2010). Cocconeis pottercovei sp. nov. and Cocconeis pinnata var. matsii var. nov., two new marine diatom taxa from King George Island, Antarctica. Diatom Research 15 (1): 1-11.

Béguinot J. (2015). Extrapolation of the species accumulation curve for incomplete species samplings: a new nonparametric approach to estimate the degree of sample completeness and decide when to stop sampling. Annual Research & Review in Biology 8: 1-9.

Chao A. & Jost L. (2012). Coverage-based rarefaction and extrapolation: standardizing samples by completeness rather than size. Ecology 93: 2533-2547.

Giesecke, T., Ammann, B., Brande A. (2014). Palynological richness en evenness: insights from the taxa accumulation curve. Vegetation History & Archaeobotany 23: 217-228.

Lopez L.C.S., de Aguiar Fracasso M.P., Oliveira Mesquita D., Torre Palma A.R. & Riul P. (2012). The relationship between percentage of singletons and sampling effort: a new approach to reduce the bias of richness estimates. Ecological Indicators 14: 164-169.

Zidarova, R., Kopalová, K., Van de Vijver, B. (2016). Diatoms from the Antarctic Region : Maritime Antarctica. Iconographia Diatomologica vol. 24. Koeltz Botanical Books, Schmitten-Oberreifenberg, Germany.

6. PLOS authors have the option to publish the peer review history of their article (what does this mean?). If published, this will include your full peer review and any attached files.

Reviewer #1: Yes: Peter Convey

Reviewer #2: No

---

## [Author Response · Author response to Decision Letter 0]

10 Sep 2019

POINT-BY-POINIT RESPONSE TO REVIEWERS

REVIEWER'S COMMENTS: 

Reviewer #1: 

Language is generally clear, though some minor syntax and spelling editing is required throughout; I note that this journal does not offer copy editing, so this will have to be done carefully during revision. Is the citation format in the text correct for the journal? It is unusual to list both numbered references and fully named author lists together.

Response to reviewer: The authors followed the guidelines for manuscript formatting for submission to Plos One Magazine. However, there were some formatting errors that have now been corrected in this review.

Abstract – the description of the role of distance (‘isolation by distance’) in structuring (or not) diatom community composition could be reworded a bit to make it clearer: at present the sentence structure seems to suggest initially that distance is being considered from each of the six sampling sites individually (for instance along a transect from a central sampling location at each), and rather it later becomes clear that the intention is to look at distance as a factor between the six sampling locations.

Response to reviewer: Its been corrected, we rewritten the description. The modification is found in the marked copy of manuscript.

Introduction – the opening para should make it clear from the outset that the focus is on diatoms; at present the opening sentences appear to make statements applying to the Antarctic flora more generally – if this is the case then they need expanding into a separate general opening para, before subsequently focussing in on diatoms specifically. However, I think the intention is to focus on diatoms from the outset.

Response to reviewer: Ok, the modification is found in the marked copy of manuscript.

Terminology in the para describing the influence of distance – it should be ‘community similarity’ rather than ‘species similarity’.

Response to reviewer: We change this, the modification is found in the marked copy of manuscript. 

Within the methods section there is no reference given to how the diatoms obtained were identified? I appreciate the long list of recent species descriptions early in the Introduction, but something specific should be said in this section. Other than these comments, the approaches and analyses used are appropriate.

Response to reviewer: Ok, we add the references used in detail, the modification is found in the marked copy of manuscript. 

Taxonomic authorities – I believe that journal format only requires these to be listed once, so there are many instances where repeat listings can be deleted. If acceptable to the journal, I would suggest placing all authorities within Table 2 and deleting them from the text, which will also help make the text more readable.

Response to reviewer: Ok, we change this, the modification is found in the marked copy of manuscript.

Diversity indices – I am not familiar with what 0D, 1D, 2D annotation means?

Response to reviewer: The explanation is at the bottom of page 8 in the following excerpt: “The alpha diversity was approximately 10 species for species richness and this analysis further showed that the true beta diversity of Shannon (1D) was lower than the richness based on species richness (0D) and based on the inverse concentration of Simpson (2D). This shows that differences between sampling sites occurred due to the contrast between the most abundant and the rarest species of each site (Table 4).”

Overall, while the study found differences in various indices between sites, community similarity was high across the entire study area, emphasised by the highest similarity being found between the most distant sites, and there was no overall relationship between separation distance and diversity. I wonder if this might reflect the overall physical scale of the study area, where most of the sampling sites were actually quite close to each other – in other words does this indicate the scale of the study was not large enough to identify such effects, and that ‘isolation’ was not a significant factor at this scale? Perhaps a larger geographical sampling scale would lead to identification of such a relationship, as predicted by the study’s overall hypothesis?

Response to reviewer: Ok, we added this discussion, the modification is found in the marked copy of manuscript.

Figures – the labels on the diatom photo figs are blurred in my pdf?

Response to reviewer: We do not understand this point. Our figures and the pdf generate in the submission process are in highest quality permited to the jounal guidelines.

Table 1 – I think I would suggest presenting this data in km rather than m

Response to reviewer: Ok, we change this, the modification is found in the marked copy of manuscript.

Reviewer #2: 

Diatom identification: In the method section of the manuscript, the authors only referenced Round et al. (1990) as the source for the identification of the diatoms they observed in their samples. This is not suitable as this book by Round et al. only deals with taxonomy at the generic level and some of the genera the authors identified where not even described at the time it was published (e.g. Psammothidium, Mayamaea). It would be much better to list all the papers and flora used for identification. In addition, I found particularly surprising that you did not make use of the flora published by Zidarova et al. (2016) specifically dealing with Maritime Antarctica, including the South Shetlands Islands. In particular, the species the authors identified as Luticola aff beyensis and Luticola muticopsis morphotype 1 may be Luticola amoena and Luticola austroatlantica, respectively, as illustrated in the book by Zidarova et al. (2016). Cocconeis aff costata may also correspond to Cocconeis pinnata var. matsii as described by Al-Handal et al. (2010) from King George Island.

Response to reviewer: Ok, we correted this, the modification is found in the marked copy of manuscript.

Spelling of diatom names: I’ve found numerous mistakes throughout the text of the manuscript, the legend of the figures and in the tables. Please pay attention to that and correct them.

Response to reviewer: Ok, we correted this, the modification is found in the marked copy of manuscript.

Rarefaction analysis: Taxonomic richness assessed through rarefaction in fact consists in an interpolation to lower (and common to all sample) count size and it results in a loss of information (Giesecke et al. 2014). Moreover, samples standardized by size can have different degrees of completeness, depending on the species-abundance distributions of the assemblages that are compared. A better way is to compare samples of equal completeness, not equal size (Chao and Jost 2012). This can be done by using the species accumulation curve to extrapolate the total species richness of the less complete samples (Béguinot 2015). Several parameters can be used to estimate the completeness of the sampling such as the percentages of species represented by one individual (=singleton) and the sample intensity, i.e. abundance divided by richness (Lopez et al. 2012).

Response to reviewer: The authors understand the suggestion, but evaluate that would not modify the results to a significant extent. Regarding the suggestion to use species represented by only one individual in a given sample to access completeness, it cannot be applied to our data, because such species are eliminated from data.

Comparison of the results: the main result is that spatial distance does not appear to drive differentiation in diatom communities in this study. Maybe this result is due to the fact that the distance covered in the study is relatively small (less than 100 km). Could you compare your result with similar studies dealing with the effect of spatial distance on diatom community composition?

Response to reviewer: Ok, we added this discussion, the modification is found in the marked copy of manuscript.

Conclusion: I think the manuscript would really gain if you could broaden a little the scope of your conclusions with a few sentences. For example, could you discuss about any wider implications regarding the potential effect of further climate warming on the resilience of this community. I read that the penguin populations were threatened as their conditions for nesting are changing (change in local weather with unprecedented rain, prematurely snow melt creating puddles of water on the ground, etc…). How would this affect Prasiola and its associated diatom flora?

Response to reviewer: Ok, we added this discussion, the modification is found in the marked copy of manuscript.

Specific comments:

Page 1: In the title, I would use: Distribution of aerophilous diatom communities … to make it clear what kind of samples were investigated.

Page 1: in the address of the authors, correct spelling of Antarctic (only one “t”)

Page 2, in abstract: rephrase such as: …such as the type of substrate…

Page 2: correct spelling of Luticola muticopsis (not multicopsis)

Page 2: add dots after initials such as: D.E. Kellogg et al.

Page 2:… in our samples, …

Page 2: …and abundance of their diatom community…

Page 2: keywords: replace “Diatom” by “aerophilous diatoms”, delete “Taxonomy” as this paper does not deal with taxonomy

Page 3: rewrite the sentence such as: The larger the distance between sites, the larger are the environmental variations and, therefore, the difference in species composition tends to be also larger. The following sentence, starting by “Spatial distance is a factor that influences…” is unclear, please re-write it.

Page 4: quantitative analyses: give the number of valves counted to obtain the relative abundance data

Page 4: the statement starting with “sample sufficiency…” is unclear

Page 4: regarding rarefaction, please see above in general comments

Page 5: separate thousands using a comma such as: 10,000

Page 5: use lower case for “also”

Page 5: …(Oksanen et al. 2016) in R.

Page 5: correct spelling of Pseudogomphonema and carlinii

Page 5: Cocconeis pinnata var. matsii is also a marine species (if it’s the correct identification?)

Page 6: correct spelling for muticopsis (several times)

Page 6: just give the authorities once, when a species is first mentioned. There is no need to repeat them every time thereafter.

Page 7: separate thousands using a comma such as: 10,000

Page 9: delete “1.” before “Hamsher”

Page 9: just use Fottea for title of the journal, delete “Czech Phycological Society – Praha, Czech Republic, 2007, currens.”

Page 9: in the reference by Kochman-Kedziora et al.: just use Fottea for title of the journal, delete “Czech Phycological Society – Praha, Czech Republic, 2007, currens.”

Page 10: delete “3.” before “Kopalova”

Page 11: use capital L for journal title: Limnetica

Page 11: delete “2.” before “Van de Vijver”

Response to reviewer: Ok, we corrected the specific comments above.

Fig. 1: add an insert showing the whole of Antarctica to help the reader immediately visualize the location of this region. Some locations on the map and the legend are in Portuguese (e.g Rei George instead of King George; Sistema de Coordenadas, etc…). Please change all to English. Also correct spelling of British. Also give explanation about the units used on sides of the figure.

Response to reviewer: Ok, we correted this, the modification is found in the marked copy of manuscript.

Figs 2 and 3: indicate to what distance the scale bar corresponds (is it 5 microns?), either directly on the figures or in the legend. In the legend: add “LM photographs”; correct spelling of diatom names, use English instead of Portuguese for morphotype

Response to reviewer: Ok, we correted this, the modification is found in the marked copy of manuscript.

Fig. 4: add “SEM photographs”, correct spelling of diatom names.

Response to reviewer: Ok, we correted this, the modification is found in the marked copy of manuscript.

Table 1: I would give the distances in kilometers instead of meters. If you insist in using meters, at least delete the decimals…

Response to reviewer: Ok, we correted this, the modification is found in the marked copy of manuscript.

Table 2: There are many mistakes in the diatom names:

Muticopsis instead of multicopsis

Cyclotella instead of Ciclotella

Fragilaria instead of Fragillaria

Hantzschia instead of Hantzchia

Carlinii instead of carlini

Pseudogomphonema instead of Pseudoghophonema

Olegsakharovii instead of olegsakharoni

Response to reviewer: Ok, we correted this, the modification is found in the marked copy of manuscript.

---

## [Decision Letter · Decision Letter 1]

4 Oct 2019

PONE-D-19-16073R1

Distribution of aerophilous diatom communities associated with terrestrial green macroalgae in the South Shetland Islands, Maritime Antarctica

PLOS ONE

Dear Dr. Victoria,

Thank you for submitting your manuscript to PLOS ONE. After careful consideration, we feel that it has merit but does not fully meet PLOS ONE’s publication criteria as it currently stands. Therefore, we invite you to submit a revised version of the manuscript that addresses the points raised during the review process.

We would appreciate receiving your revised manuscript by Nov 18 2019 11:59PM. To enhance the reproducibility of your results, we recommend that if applicable you deposit your laboratory protocols in protocols.io, where a protocol can be assigned its own identifier (DOI) such that it can be cited independently in the future. For instructions see: http://journals.plos.org/plosone/s/submission-guidelines#loc-laboratory-protocols

We look forward to receiving your revised manuscript.

Kind regards,

Jinzhuang Xue

Academic Editor

PLOS ONE

Additional Editor Comments (if provided):

The reviewer provided two issues, which have been raised during the first round of reviewing but not been well dealt with during the revision. I agree with the comments of the reviewer. 1. Please consider more about the roles of rare species in the estimation of species richness. There are many methods in ecology to deal with rare species, rather than simply deleting them from the original data. 2. Please consider the comments about distance. 3. Please carefully check the text to kill the typos, some of which have been marked in the attached file from the reviewer.

Reviewers' comments:

Reviewer's Responses to Questions

**Comments to the Author**

1. If the authors have adequately addressed your comments raised in a previous round of review and you feel that this manuscript is now acceptable for publication, you may indicate that here to bypass the “Comments to the Author” section, enter your conflict of interest statement in the “Confidential to Editor” section, and submit your "Accept" recommendation.

Reviewer #2: (No Response)

2. Is the manuscript technically sound, and do the data support the conclusions?

Reviewer #2: Partly

3. Has the statistical analysis been performed appropriately and rigorously? 

Reviewer #2: No

4. Have the authors made all data underlying the findings in their manuscript fully available?

Reviewer #2: (No Response)

5. Is the manuscript presented in an intelligible fashion and written in standard English?

Reviewer #2: Yes

6. Review Comments to the Author

Reviewer #2: Review of manuscript PONE-D-19-16073R1 by Da Silva et al.

With this revised version the authors have improved their manuscript substantially.

There are however two issues with the reply the authors gave to my comments:

1. “Regarding the suggestion to use species represented by only one individual in a given sample to access completeness, it cannot be applied to our data, because such species are eliminated from data.“ I really don't understand this reply. Species richness (i.e. the number of species present in the sample) is more or less the basis of this study, so why would you "eliminate" the rare species from the data? Please explain? In any case there was no mention of this “elimination” in the method section of the paper.

2. I also asked the authors to consider in their discussion that the relatively small distance (less than 100 km) covered in this study may in part explains the lack of relationship between spatial distance and the diatom communities similarity. Note that the other reviewer made the same remark. The authors did not answer this question directly and I could not found any mention of it in the revised version of the manuscript.

In addition, there are still a few typos in the manuscript. Please see the PDF with my comments as “sticky notes”.

Overall, I think this study can be published after minor revision.

7. PLOS authors have the option to publish the peer review history of their article (what does this mean?). If published, this will include your full peer review and any attached files.

Reviewer #2: No

---

## [Author Response · Author response to Decision Letter 1]

2 Dec 2019

Dear Editor and Reviewers,

We would like to inform you that image 4 of the manuscript "Distribution of aerophilous diatom communities associated with terrestrial green macroalgae in the South Shetland Islands, Maritime Antarctica", with manuscript number PONE-D-19-16073R1 submitted to the journal Plos One, has not been authorized for disclosure or publication. We have therefore removed it from the updated version of the article. Nevertheless we would like to emphasize that the lack of this image does not change the context of the paper, nor does it, in any way, impair the results described, as the species previously shown in image 4 are also represented in Figures 2 and 3 as follows:

1. Cocconeis pinnata var. matsii – Cited in Figure 2 (4-10)

2. Luticola amoena - Cited in Figure 2 (14-17)

3. Luticola multicopsis - Cited in Figure 2 (22-25)

4. Navicula aff perminuta - Cited in Figure 3 (28-31)

5. Pinnularia austroshetlandica - Cited in Figure 3 (32-34)

6. Psammothidium germainii - Cited in Figure 3 (45-48)

7. Pseudogomphonema kamtschaticum - Cited in Figure 3 (51-55)

8. Pteroncola carlinii - Cited in Figure 3 (56-59)

In response to questions:

Editor Comments:

“1. Please consider more about the roles of rare species in the estimation of species richness. There are many methods in ecology to deal with rare species, rather than simply deleting them from the original data.”

Regarding the suggestion to use species represented by only one individual in a given sample to access completeness, we argue that in such cases we might be examining the result of a contaminant. Species represented by only one valve in each sample were excluded from the analyses due to the difficulty of a correct species determination and to avoid potential contaminants. We have added this fact to the manuscript methodology along with references.

“2. Please consider the comments about distance.” 

We added a discussion about it in the manuscript.

“3. Please carefully check the text to kill the typos, some of which have been marked in the attached file from the reviewer.”

The text were revised.

---

## [Editor Report · Decision Letter 2]

5 Dec 2019

Distribution of aerophilous diatom communities associated with terrestrial green macroalgae in the South Shetland Islands, Maritime Antarctica

PONE-D-19-16073R2

Dear Dr. Victoria,

We are pleased to inform you that your manuscript has been judged scientifically suitable for publication and will be formally accepted for publication once it complies with all outstanding technical requirements.

With kind regards,

Jinzhuang Xue

Academic Editor

PLOS ONE
---

## [Editor Report · Acceptance letter]

18 Dec 2019

PONE-D-19-16073R2 

Distribution of aerophilous diatom communities associated with terrestrial green macroalgae in the South Shetland Islands, Maritime Antarctica 

Dear Dr. Victoria:

I am pleased to inform you that your manuscript has been deemed suitable for publication in PLOS ONE. Congratulations! Your manuscript is now with our production department. 

With kind regards,

on behalf of

Dr. Jinzhuang Xue 

Academic Editor

PLOS ONE